# A Special Network Comprised of Macrophages, Epithelial Cells, and Gut Microbiota for Gut Homeostasis

**DOI:** 10.3390/cells11020307

**Published:** 2022-01-17

**Authors:** Wei Chen, Dan Liu, Changhao Ren, Xiaomin Su, Chun-Kwok Wong, Rongcun Yang

**Affiliations:** 1Department of Immunology, School of Medicine, Nankai University, Tianjin 300071, China; cwnk@mail.nankai.edu.cn (W.C.); 2120211445@mail.nankai.edu.cn (D.L.); changhaoren@mail.nankai.edu.cn (C.R.); xiaominsu@nankai.edu.cn (X.S.); 2Department of Chemical Pathology, The Chinese University of Hong Kong, Hong Kong 999077, China; ck-wong@cuhk.edu.hk; 3Translational Medicine Institute, Affiliated Tianjin Union Medical Center, Nankai University, Tianjin 300071, China; 4State Key Laboratory of Medicinal Chemical Biology, Nankai University, Tianjin 300071, China

**Keywords:** gut macrophage, differentiation, tolerance, epithelial barrier, bacteria recognition, immunomodulation, proliferation

## Abstract

A number of gut epithelial cells derived immunological factors such as cytokines and chemokines, which are stimulated by the gut microbiota, can regulate host immune responses to maintain a well-balance between gut microbes and host immune system. Multiple specialized immune cell populations, such as macrophages, dendritic cells (DCs), innate lymphoid cells, and T regulatory (Treg) cells, can communicate with intestinal epithelial cells (IEC) and/or the gut microbiota bi-directionally. The gut microbiota contributes to the differentiation and function of resident macrophages. Situated at the interface between the gut commensals and macrophages, the gut epithelium is crucial for gut homeostasis in microbial recognition, signaling transformation, and immune interactions, apart from being a physical barrier. Thus, three distinct but interactive components—macrophages, microbiota, and IEC—can form a network for the delicate and dynamic regulation of intestinal homeostasis. In this review, we will discuss the crucial features of gut microbiota, macrophages, and IEC. We will also summarize recent advances in understanding the cooperative and dynamic interactions among the gut microbiota, gut macrophages, and IEC, which constitute a special network for gut homeostasis.

## 1. Introduction

The gut microbiota contains bacteria, fungi, archaea, and protozoa. They play a critical role in human health and diseases [1]. The most abundant phyla in the human intestine are *Firmicutes*, *Bacteroidetes*, *Proteobacteria*, and *Actinobacteria* [2]. Commensal bacteria promote intestinal immune system maturity [3] and protect against pathogen colonization [4]. Lactic acid bacteria (*Firmicutes* phylum) are marked as probiotic bacteria that are beneficial for human health. Despite the commensal benefits of the gut microbiota, the presence of aberrant microbes or perturbation of the gut microbiota can lead to intestinal disorder and inflammation [5]. *Escherichia coli* (*E. coli*) can cause infection or diseases under certain conditions. These aberrant microorganisms may be present within a healthy microbiota. The microbiota commensal microbes can control the expansion of these microorganisms. These gut microbiota can be altered by host genetics, overuse of antibiotics, and changes in diet [6].

Tissue resident macrophages are highly heterogeneous. Each subset of macrophages possesses a unique transcriptome to fulfill niche-specific functions [7,8]. Gut macrophages are strategically underlying the intestinal epithelium, mostly in the lamina propria (LP) [9,10]. The macrophages in LP contribute to host defense and barrier integrity, as well as the constitutive secretion of interleukin (IL)-10 for the maintenance of FoxP3^+^ T regulatory cells (Treg) [11]. The gut macrophages can be continually replenished by bone marrow derived monocytes in a chemokine receptor 2 (CCR2) dependent manner [12]. These gut macrophages are also derived from embryonic progenitors in the yolk sac and/or fetal liver [13].

The gut epithelium is composed of a single layer of cells, thereby forming crypt villus units [14]. This single layer epithelium alongside mucus segregates the core body from the luminal contents, providing a physical barrier. Furthermore, intestinal epithelial cells (IEC) secrete antimicrobial peptides (AMPs) and cytokines to coordinate the regulation of the active immune responses to infection, inflammation, and homeostasis [15]. In general, the epithelium consists of stem cells, absorptive enterocytes, Paneth cells, goblet cells, and enterochromaffin cells [16]. However, the mucus content and composition correlate with the composition of the gut microbiota in the small and large bowel, which is markedly different in both gut sections.

In the intestinal mucosal environment, gut epithelial cell-derived factors, including cytokines such as IL-18 and chemokines, which are stimulated by the gut microbiota, modulate host immune responses to maintain a good balance between the gut microbes and the host immune system. However, multiple specialized immune cell populations such as macrophages, innate lymphoid cells, dendritic cells (DCs), and Treg cells can communicate with the IEC and/or the gut microbiota bidirectionally [17,18]. The luminal content, such as commensal microbes, can also be sampled by mucosal macrophages. Thus, three distinct but interactive components—gut microbiota, immune cells such as macrophages, and IEC—can form a special network for the delicate and dynamic regulation of intestinal homeostasis (Figure 1). In this review, we will discuss crucial features of the gut microbiota, macrophages, and IEC. We will also summarize recent advances in understanding the cooperative and dynamic interactions of the network comprised of the gut microbiota, macrophages, and IEC.

## 2. Macrophages and Gut Microbiota

Intestinal macrophages reside either within the LP or the muscle layer. Muller et al. [9] discussed recent advances regarding the origin, phenotype, and function of the macrophages residing in the different layers of the intestine during homeostasis. LP macrophages (LPMs) can be further subdivided into mucosal and submucosal LPMs [19]. Mucosal LPMs can line the intestinal epithelium and vasculature in the murine small and large intestines [20,21]. They contribute to the host defense and barrier integrity, as well as interleukin (IL)-10, which promotes the maintenance of FoxP3^+^ Treg [22]. Perivascular macrophages in the small intestine and colon mainly participate in the regulation of the vasculature [19,21]. Submucosal LPMs can sense luminal antigens to protect the mucosa from enteropathogens, whereas macrophages residing in the muscularis are essential for tissue homeostasis. The interaction between muscularis macrophage and neurons controls intestinal motility and protects gut tissues during injury and infection. However, it also creates tissue damage in gastrointestinal disorders, such as gastroparesis [23]. These subsets of macrophages with different transcriptional profiles are further confirmed through single-cell RNA sequencing technologies [24,25,26,27] (Figure 2). There is a “monocyte waterfall” from circulation to the intestine in order to maintain the macrophage pool in the gut in a CCR2 dependent manner in the murine colon [12]. These monocytes are identified as the ly6c^hi^ CX3CR1^int^ MHCII^−^ subset and exhibit pro-inflammatory properties as they act in the circulation. In the resting intestine, they terminally become mature resident ly6c^low/−^ CX3CR1^hi^ MHCII^hi^ macrophages, which express IL-10 and maintain intestinal homeostasis. These macrophages are the main cell source for the tolerogenic IL-10. In addition, two intrinsic sources also exist, embryo derived F4/80^hi^ CD11b^low^ macrophages and yolk sac derived macrophages, in the colon or small intestine [12,13]. Fate mapping and single cell RNA sequences have identified these macrophages that arise from both embryonic precursors, which express different transcriptional profiles in the murine small intestine [19].

### 2.1. Gut Microbiota and Resident Macrophages

#### 2.1.1. Effects of Gut Microbiota on the Resident Macrophages

Gut microbes are essential for the differentiation and function of resident macrophages. In germ-free (GF) mice, the number, phenotype, and function of the gut macrophages are impaired; but they can be restored by microbial colonization from adult mice [28]. IL-1β, IL-12, and IL-10 expressions in small intestinal macrophages are reduced in microbial depleted conditions [29,30,31]. The commensal microbiota can promote the development of two major subsets, CD11c^+^CD121b^+^ and CD11c^−^ CD206^hi^, resident macrophages in the colon [25]. The altered intestinal microbiota can potentially cause a shift between inflammatory macrophages to tolerant macrophages in the colon or small intestine [32,33,34]. High-fat diet-induced dysbiosis mediates macrophage chemotactic protein (MCP)-1/CCR2 axis-dependent M2 macrophage polarization in the murine small intestine [34]. Notably, several bacteria strains, which can induce differentiation and function of resident macrophages in the colon, have been identified, such as *Enterococcus faecalis* (*E. faecalis*), *Streptococcus gallolyticus* (*S. gallolyticus*), and *E. coli* [35,36]. Fusobacterium *nucleatum* (*F. nucleatum*) also triggers macrophage activation through TLR4 and subsequent IL-6/STAT3/c-MYC signaling, supporting its anti-inflammatory polarization in the colon [37].

#### 2.1.2. Effects of Gut Microbiota Metabolites on Resident Macrophages

(1) Short-chain fatty acids (SCFAs), including acetate, propionate, and butyrate, are the major metabolic products of the gut microbiota digestion of non-absorbable dietary fiber and resistant starches [38]. Acetate is produced from pyruvate in two different ways, acetyl-CoA by enteric bacteria and Wood−Ljungdahl by acetogens, whereas propionate can be produced by the lactate pathway by *Firmicutes* and the succinate pathway by *Bacteroidetes* [39]. *Eubacterium*, *Clostridia*, *Ruminococcaceae*, and *Firmicutes* are the main producers of butyrate from Acetyl-CoA [39]. These SCFAs play an important role in inducing resident macrophages in the colon [40,41,42]. Treatment with the butyrate directly causes metabolic reprogramming of the intestinal macrophages and prevents macrophage from dysfunction [42]. SCFAs exert anti-inflammatory effects through binding to G-protein-coupled receptor 43 (GPR43) [43] and GPR109a expressed in the macrophages [44]. The butyrate can act as a histone deacetylase (HDAC) inhibitor [45] and directly suppress NF-κB activation to regulate the intestinal macrophage function in the colon [46], exerting its anti-inflammatory effects. In addition, butyrate also inhibits HDAC3 to reduce the activation of the mammalian target of rapamycin (mTOR) and glycolysis, which contribute to the antimicrobial function of macrophages [47].

(2) Tryptophan (TRP) metabolites can be metabolized into indole, indole-3-acrylic acid (IA), indole-3-acetic acid (IAA), indole-3-carboxalaldehyde (ICAld), indole-3-propionic acid (IPA), indole-3-lactic acid (ILA), indole-3-acetonitrile (IACN), indole-3-ethanol (IE), and indole-3-carboxylic acid (ICA) [48,49]. Some bacterial species such as *Proteus vulgaris* can convert TRP into indole by using tryptophanase [50]. *Clostridium sporogenes* and *Peptostreptococcus* spp. produce IA [51,52]. *Clostridium sporogenes* and *Rominococcus gnavus* produce tryptamine [53]. Recent evidence has shown that inflammation is related with the alterations in the metabolism of TRP. Kynurenine (Kyn)/Trp is also used as an indicator of the progression of inflammation [54]. Notably, both metabolites tryptamine and indole-3-acetate (I3A) can reduce the fatty-acid- and LPS-stimulated production of pro-inflammatory cytokines in the macrophages in the colon [55].

(3) Secondary bile acids. Primary bile acids are synthesized by hepatocytes. Approximately 5% of bile acids are converted into secondary bile acids by microbiota species in the cecum and colon. Lithocholic acid (LCA) and deoxycholate (DCA) are the two major types generated through 7a-dehydroxylation of cholic acid (CA) and chenodeoxycholic acid (CDCA), respectively. *Bacteroides*, *Bifidobacteria*, and *Lactobacillus* can synthesize secondary bile acids from primary bile acids [56,57]. Interestingly, the bile acid-activated receptors (FXR, vitamin D receptor (VDR), liver X receptor (LXR), GPBAR1, and pregnane X receptor (PXR)) can be detected in myeloid cells [58]. Furthermore, the binding of the bioactive molecules to receptors such as the G protein-coupled bile acid receptor 1 (GPBAR1 or Takeda G-protein receptor 5(TGR5)) and farnesoid-X-Receptor (FXR) on the macrophages may exert an anti-inflammatory role [59,60]. TGR5 also regulates the M1/M2 phenotype of the intestinal macrophages [61]. Bile acid inhibits LPS mediated proinflammatory cytokines in a TGR5-dependent manner in the macrophages [62]. The knockout of TGR5 in mice can accelerate LPS-mediated inflammation in the liver and abolish the suppressive effect of the TGR5 agonist on inflammatory cytokines [63]. A TGR5-specific semisynthetic bile acid suppresses the foam cell formation of macrophages by decreasing the uptake of low dense lipoprotein (LDL) [64].

(4) Other metabolites. Branched-chain amino acids (BCAAs) can inhibit LPS-induced nitric oxide (NO) and IL-6, and decrease the damage by H_2_O_2_ in macrophages in vitro [65]. Symbiotic polyamine can cause increased anti-inflammatory macrophages in the colon [66]. Quercetin-3-glucuronide inhibits the expression of scavenger receptor class A type 1 (SR-A1) and CD36, and also suppresses the formation of foam cells in RAW 264.7 cells [67]. Recent studies have reported that microbial metabolites UroA/UAS03 and polysaccharides can promote IL-10 production and mediate anti-inflammatory activities in colon macrophages [68,69]. Moreover, Ifrim et al. also reported that high doses of pattern recognition receptors (PRR) ligands such as flagellin, lipopolysaccharide, and poly I:C, can led to a tolerance of macrophages [70].

#### 2.1.3. Effects of Gut Microbiota on the Other Resident Macrophages

Recent studies have also found the effects of the gut microbiota on perivascular macrophages and muscularis macrophages. Honda et al. [21] demonstrated that a tight anatomical barrier between microbiome and LP perivascular macrophages can prevent bacterial translocation. Gut microbiota alternation with age can change the phenotype of muscularis macrophages (MMs) and affect gastrointestinal motility [71]. Recent data have shown that MMs upregulate neuroprotective factors via β_2_-adrenergic receptor (β_2_-AR) signaling [72]. The roles of the gut microbiota in the association between gastrointestinal motility and 5-hydroxytryptophan (HT) expression have also been found in resident macrophages of the gastrointestinal tract [73]. However, long-living tissue-resident macrophages, which express both CD4 and TIM4 molecules, are not affected by the microbiota in adult mice [13].

### 2.2. Effects of Gut Resident Macrophages on Microbiota

Macrophages are one of the critical cell populations for maintaining intestinal homeostasis [74]. Macrophage depletion using clodronate liposomes can alter the composition of the gut microbiota in the azoxymethane (AOM)/dextran sulfate sodium (DSS) mouse model of colon cancer [75]. In the macrophage-depleted group, increased *Firmicutes* has been found [75]. Recently, Earley et al. [76] found that intestinal macrophages are important in the normal colonization of gut microbes in adult zebrafish via macrophage-deficient interferon regulatory factor 8 (IRF8).

## 3. Gut Epithelium and Microbiota

Gut epithelium stands as a single-cell barrier between the intestinal microbiota and the submucosal immune cells, representing the primary shield to prevent microbial translocation. It is formed by various differentiated cell types, including stem cells, enterocytes, goblet cells, Paneth cells, Tuft cells, and M cells [77]. These cells can potentially develop specialized functions such as intestinal enteroendocrine cells that comprise at least eight cellular subsets, including enterochromaffin cells (5-HT/serotonin), D cells (somatostatin), and G cells (gastrin) [78].

### 3.1. Gut Epithelial Structure and Function Needs Gut Microbiota

#### 3.1.1. Effects of Gut Microbiota on the Gut Epithelial Cells

Gut epithelial structure and function needs in the gut microbiota. 

*Lactobacillus* spp., a kind of generally recognized probiotic, can promote the proliferation of intestinal epithelial cells (IEC) by nicotinamide adenine dinucleotide phosphate (NADPH) oxidase-1 dependent reactive oxygen species [79]. Compared with healthy volunteers, *F. prausnitzii* in patients with inflammatory bowel disease and irritable bowel syndrome is reduced [80,81,82]. Recent studies have shown that *F. prausnitzii* plays a protective role in the intestinal tissue [83,84,85].

#### 3.1.2. Effects of the Commensal-Derived Metabolites on Gut Epithelial Cells

(1) SCFAs. SCFAs can modulate the proliferation, differentiation, and functions of IEC to impact gut motility and gut barrier functions, as well as host metabolism [86]. Luminal butyrate can serve as a source of energy for gut epithelial cells in metabolic processes, and act to inhibit HDAC activity in intestinal epithelial stem cells [87,88]. SCFAs also affect pro-inflammatory cytokine production through enhancing NF-κB activation in epithelial cells [89]. Goblet cell differentiation can be influenced by acetate, a SCFA produced by *Bifidobacterium* in a gnotobiotic rodent model [90]. Considerable evidence supports that the concentration of SCFAs is decreased in the colonic lumen of ulcerative colitis (UC) [91]. The dybiosis, which is characterized by decreased butyrate-producing species *Eaecalibacterium prausnitizii* and *Roseburia hominis*, is also defined in patients with UC [80]. SCFAs-sensing GPCRs can maintain intestinal epithelial barrier homeostasis via protecting against intestine inflammation. Studies have also found that G protein-coupled receptor (GPR)109a^−/−^ and GPR43^−/−^ mice could suffer from more severe DSS-induced colitis [44]. The administration of SCFAs in UC patients can ameliorate colitis [92]. The SCFA mixture (sodium acetate, sodium propionate, and sodium butyrate) enhances the efficacy of IBD treatments [93].

(2) TRP metabolites. Studies have found that indole derivatives of TRP catabolism can directly stimulate human and mouse Trpa1 and intestinal 5-HT secretion to increase intestinal motility [94]. TRP metabolites indole-3-ethanol, indole-3-pyruvate, and indole-3-aldehyde can impede gut permeability by maintaining the integrity of the apical junctional complex and the associated actin regulatory proteins [95]. TRP also alleviates inflammatory responses to the ameliorate barrier integrity through the CaSR/Rac1/PLC-γ1 signaling pathway to enterotoxigenic *Escherichia coli* in porcine intestinal epithelial cells [96]. Studies have also found that transepithelial electrical resistance can be increased by indole-3-propionic acid (IPA). IPA also decreases paracellular permeability via the increase in tight junction proteins (claudin-1, occludin, and ZO-1) [97].

(3) Secondary bile acids. The secondary bile acid receptor, TGR5, is expressed in the entire gastrointestinal tract. This receptor regulates a lot of gut homeostatic functions [98,99]. For example, endogenous bile acids in the intestinal lumen can promote intestinal stem cell (ISC) renewal and drive regeneration in response to injury [100]. Notably, bile acid-activated receptors such as FXR, GPBAR1, RORγt, VDR, and PXR have a close relationship with IBD [101]. Studies have found that altered bile acid-activated receptors can be found in patients with IBD. Thus, in these patients with IBD, restoring bile acid signaling might be beneficial [101]. Studies have indicated that the TGR5 ligand can exert profound effects in rodent models of colitis [61].

(4) Other metabolites. Other commensal-derived metabolites are also involved in preserving gut epithelial structure and function, such as polyamines [102]. Epithelial proliferation and macrophage differentiation could be regulated by symbiotic polyamine in the colon [66]. Microbe-derived lactate also is a potent inducer of colonic hyper-proliferation in mice [103]. The microbial metabolites can maintain the homeostasis of gut stem cells via the Wnt/β-catenin pathway [104]. Other signaling pathways, such as the JAK/STAT pathway, are also necessary in the bacteria-modulated epithelium homeostasis [105,106,107]. However, the mechanisms involved in these regulations are too complicated to make a complete interpretation.

#### 3.1.3. Effects of Gut Microbiota on the Gut Organoids

Studies of intestinal epithelial organoids also support that the gut epithelial structure and function needs the gut microbiota. Cryptosporidium parvum infection attenuates ex vivo propagation of murine intestinal enteroids [108]. Lactobacillus accelerates intestinal stem cell regeneration via activation of the signal transducer and activation of the transcription 3 (STAT3) signaling pathway [109]. *Akkermansia muciniphila* (*A. muciniphila*), an abundant member of the human gut microbiota can ferment mucus glycoproteins into propionate and acetate in the mucus layer [110]. After exposure to *A. muciniphila*, mouse gut organoids change the expression of host transcription factors, indicating the effects of *A. muciniphila* on metabolic activity [111].

### 3.2. Effects of Gut Epithelial Cells on Gut Microbiota

Gut epithelial cells can produce AMPs, including defensins, C-type lectins such as regenerating islet-derived protein Ⅲγ (Reg IIIγ), and lysozyme [112]. These AMPs target conservative and essential bacterial features and structures. For instance, pore-forming defensins are aimed at the surface membrane, whereas C-type lectins have a strong effect on Gram-positive cell wall peptidoglycans. The antibacterial lectin RegIIIγ promotes the spatial segregation of the microbiota and host in the intestine [113]. Chemokines are shown to possess an antimicrobial activity, including Gram positive and Gram negative bacterial pathogens [114], such as that cys-x-cys ligand 9 (CXCL9) contributes to antimicrobial protection of the gut during *Citrobacter rodentium* infection, independent of chemokine-receptor signaling [115]. The necleotide-binding oligomerization domian, leucine rich repeat, and pyrin domain conyaining proteins (NLRP9b), which are specifically expressed in intestinal epithelial cells [116], restrict rotavirus infection in intestinal epithelial cells. In addition, inflammasomes can regulate the gut microbiota composition in mice models. The absence of inflammasome components is also related to pathologic alterations in the gut microbiota or dysbiosis [117]. Immunoglobulins A, which is secreted into the lumen, offers immune protection by binding the bacteria or viruses and decreasing their invasion [118]. A recent study identified a highly glycosylated glycosylphosphatidylinositol (GPI)-anchored protein called Ly6/Plaur domain-containing 8 (Lypd8), which contributes to the segregation of intestinal bacteria and intestinal epithelia in the large intestine [119]. Tuft cells, taste-chemosensory epithelial cells, can eliminate parasites by producing IL-25 [120]. These strategies help to keep a balanced microbial quantity and composition [121].

## 4. Gut Resident Macrophages and Gut Epithelial Cells

Intestinal resident macrophages and IEC not only are geographically close to but also communicate and interact closely with each other. Macrophages act as innate immunological effector cells, and many of their intrinsic responses are bound up with the interpretation of microbiota-derived signals by IEC. A large number of resident macrophages are located close to the epithelial layer and considered to sample and present luminal antigens, phagocytize dead cells and work for a harmonious gut environment [9]. Tissue-resident macrophages can translocate sampled lumenal components to CD103^+^ tolerogenic dendritic cells, which in turn migrate to the mesenteric lymph nodes to induce Tregs. Recently, CX3CR1^+^ macrophages have been discovered to directly migrate across the epithelium through paracellular channels [122]. It is also possible that intestinal macrophages uptake IEC to obtain antigens indirectly. This was found in follicle-associated epithelia, which uptake and deliver antigens from the lumen to antigen-presenting cells for the induction of antigen-specific IgA [123]. Notably, gut epithelial cells can also condition macrophages towards the resident phenotype. Furthermore, gut epithelial structure and function also need gut macrophages.

### 4.1. Gut Epithelial Cells Condition Macrophages towards Resident Phenotype

IEC can condition macrophages towards a tolerogenic phenotype. They control macrophage differentiation by thymic stromal lymphopoietin (TSLP), transforming growth factor-β (TGF-β) and prostaglandinE-2 (PGE-2) [124,125]. Macrophage differentiation can also be regulated by colony-stimulating factor (CSF1). The macrophages are deficient in the mice lacking CAF1 factor (Csf1op/op) or the receptor, CSF1R [126]. The IL-10–IL-10R axis plays a crucial role in conditioning the behavior of macrophages and maintains a non-inflammatory state of macrophages in the mucosa. Gut homeostasis is also supported by the research from humans with function mutations in IL-10R [127]. In the excessive inflammation, semaphoring 7A expressed on the basolateral membrane of IEC is capable to induce IL-10 production by intestinal resident macrophages and ameliorate inflammation in DSS-induced colitis [128]. In the gut lumen, the presence of commensals promotes IEC-derived IL-25 secretion, which reduces IL-23 secretion by macrophage [129]. Converging evidence from human observational studies, population genetics, and studies in mice supports the importance of IL-23 in the mucosal inflammation in the gut in particular [130]. Two intestinal macrophage subsets, CD103^+^CD11b^+^ and CD103^−^CD11b^+^, are able to sample apoptotic IEC [131]. Although the specific genes and pathways are different among macrophages, a common “suppression of inflammation” signature has been noted [131]. These data address the role of IEC in immunosuppression and homeostatic regulation partially through intestinal macrophages. Studies have also found that local tissue-derived signals can control the functional polarization of resident macrophages [132].

### 4.2. Gut Epithelial Structure and Function Needs Gut Resident Macrophages

Studies have found that macrophages can worsen tissue injury via producing reactive oxygen species and other toxic mediators. However, they also produce a variety of growth factors to regulate epithelial and endothelial cell proliferation. Macrophages can influence the permeability of the epithelium barrier through IL-6 and NO [133]. Activated macrophages supported tissue repair by up-regulating the expression of IL-3 and IL-4 in a murine model of acute epithelial regeneration in the colon [134]. Macrophages associated with the crypts of Lieberkuehn in the colon can help the proliferation and survival of epithelial progenitor cells [135]. Yeh and colleagues [136] showed that the interaction of bone marrow derived macrophages with intestinal stromal cells is critical for the development of radiation-induced fibrosis in mouse models. After depleting CSF1R-dependent macrophages, Paneth cell differentiation is impaired and the population of Lgr5^+^ intestinal stem cells is reduced, which further impede goblet cell density and the development of M cells in Peyer’s patches in the murine small intestine [137]. Wnt proteins, which are essential for intestinal epithelial proliferation, can be secreted directly by macrophage derived extracellular vesicles [138]. Depletion of extracellular vesicles from the macrophage conditioned medium can rescue ISCs from radiation lethality [138]. Meanwhile, intestinal macrophages may induce the secretion of the Wnt1 inducible signaling protein 1 in mice [139]. IL-10 produced by macrophages also plays a critical role in recovery from intestinal epithelial injury in animal models [140]. Macrophage-derived IL-10 can result in epithelial cAMP response element-binding protein (CREB) activation and in the production of pro-repair WNT1-inducible signaling protein 1 (WISP-1), which induces epithelial cell proliferation and wound closure [140]. In addition, macrophages are able to induce IL-22 production by innate lymphoid cells (ILC)3, which in turn promotes pSTAT3 signaling in IEC and protects them from intestinal injury in the murine colon [141,142]. IFN also increases the proportion of CCR2-dependent macrophages and interleukin (IL)-22-producing innate lymphoid cells via acting on the intestinal epithelial cells. During infection with the helminth *Heligmosomoides polygyrus bakeri*, antibodies and helminth induce macrophages to produce Cxcl2 for myofibroblasts to carry out normal tissue repair in the small intestine [143]. In addition, IL-36α, which is mainly expressed by CD14^+^CD64^+^ inflammatory macrophages, also contributes to colonic epithelial cell proliferation [144].

## 5. Conclusions

As discussed above, gut macrophages, epithelium, and microbiota cooperate for inflammation and tolerance, forming a special mucosal network for gut homeostasis. Nevertheless, many aspects of cooperation among the gut macrophages, epithelium, and microbiota remain poorly understood. It will be necessary to further determine how changeable microenvironments, including abundant commensals, different niches, and immune status, give rise to diverse differentiation and functions of the gut macrophages. With advanced technology appearing in recent years, such as single-cell RNA sequencing, we will be able to reveal the macrophage landscape more precisely and comprehensively. IEC can segregate gut microbes and gut immune cells via mucosal barriers, sense signals from both populations, and secrete humoral factors, which maintain gut homeostasis. However, how these gut epithelial cells perform their tasks in different pathogen invasions, susceptible and pathological states, and over-reactive or suppressive immune responses needs to be further understood. The probiotic microbes and immune cells like macrophages support typical epithelial structure and functions, where the exact mechanisms are not completely understood. Given the situation that many gut diseases are associated with aberrant microbiota, impaired epithelium, and inappropriate immune response, more detailed knowledge is required to facilitate new diagnoses and therapies for inflammatory bowel diseases and other relative diseases.

## Figures and Tables

**Figure 1 cells-11-00307-f001:**
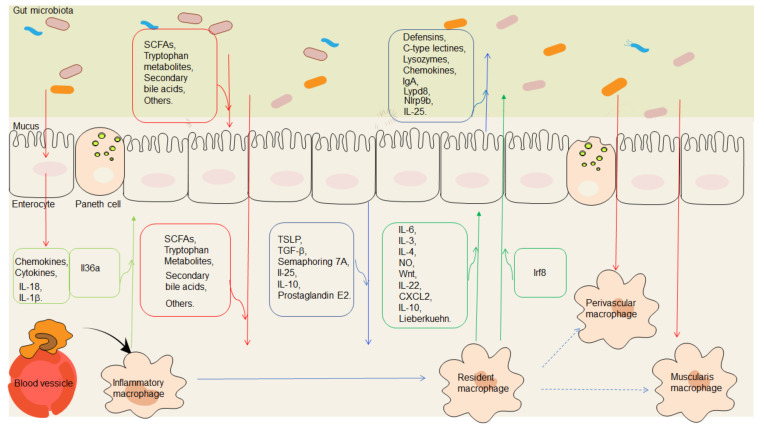
A special network comprised of macrophages, the gut microbiota, and epithelial cells for gut homeostasis. Immunological mediators, including cytokines and chemokines secreted from the gut epithelial cells stimulated by gut microbiota, such as IL-18 and IL-1β, modulate host immune responses and maintain a well-balanced relationship between gut microbes and the host immune system. The metabolites of the gut microbiota, such as short-chain fatty acids (SCFAs), tryptophan metabolites, secondary bile acids, and polyamines, regulate the proliferation and function of the gut epithelial cells. The metabolites of the gut microbiota, such as SCFAs, tryptophan metabolites, secondary bile acids, UroA/UAS03, polysaccharide, and polyamine can promote the differentiation of macrophages into resident macrophages. Substances such as defensins, c-type lectins, lysozymes, chemokines, Ly6/Plaur domain-containing 8 (Lypd8), Nlrp9b, and interleukin (IL)-25, produced by the gut epithelial cells, especially Paneth cells, also have effects on the gut microbiota. Intestinal epithelial cells (IEC) also produce factors such as thymic stromal lymphopoietin (TSLP), TGFβ, semaphoring 7A, transforming growth factor (TGF-β), retinoic acid, IL-25, and apoptotic cells to promote the macrophages into resident macrophages. Conversely, macrophages can generate some factors such as IL-6, IL-3, IL-4, nitric oxide (NO), Wnt, IL-10, and Lieberkuehn to regulate the proliferation and function of the gut epithelial cells. Meanwhile, macrophages can directly or indirectly produce effects on the gut microbiota. Recent studies have also found effects of the gut microbiota on the perivascular macrophages and muscularis macrophages. Red lines with arrows indicate that the effects of the gut microbiota or their metabolites in the gut contents on the macrophages or gut epithelial cells. Blue lines with arrows indicate that effects of the gut epithelial cell derived factors on the macrophages or gut microbiota. Green lines with arrows indicate the effects of the macrophage derived factors on the gut epithelial cells or gut microbiota. Boxes indicate the components from the gut microbiota, gut epithelial cells, or macrophages.

**Figure 2 cells-11-00307-f002:**
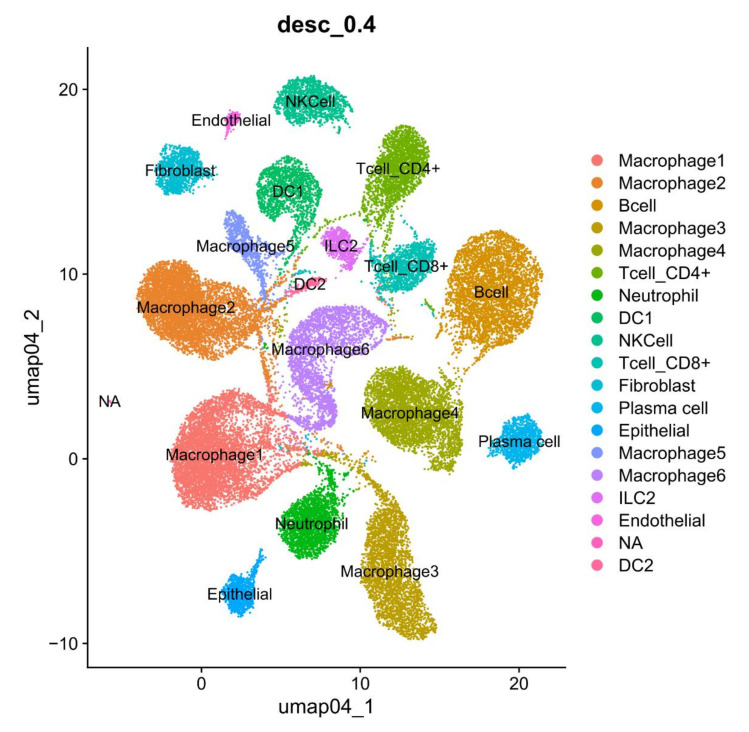
Subpopulations of macrophages in the colon tissues by scRNA-seq analyses. CD45^+^CD11b^+^ cells in the colon tissues were sorted and then used as scRNA-seq analyses.

## Data Availability

Not applicable.

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
