# Peer review of "A Special Network Comprised of Macrophages, Epithelial Cells, and Gut Microbiota for Gut Homeostasis"

_cells, 2022, doi:10.3390/cells11020307_

Round 1
Reviewer 1 Report
This manuscript discusses an interesting topic, such as the relationship between intestinal microbiota, epithelial cells and the immune system, with particular attention to macrophages which represent the most abundant immune population at the LP of the colon. However, some parts of the manuscript need improvement. In particular, paragraph 2 in which the authors discuss the relationship between macrophages and microbiota should be considerably expanded.
In the discussion of this interesting topic, the authors cite various articles that also affirm partly conflicting theses sequentially confusing the reader especially if a little inexperienced. The authors should reformulate this paragraph in order to make it clearer by separating the articles concerning the anti-inflammatory effect exerted by the microbiota from those concerning the pro-inflammatory effect of the microbiota.
Furthermore, in the part concerning secondary bile acids produced by the microbiota in the literature there are now many studies also concerning specifically the polarization of macrophages and a review on this topic was published recently in this journal. Moreover, bile acid receptors are emerging as an interesting therapeutic target in IBD, being expressed both in epithelial cells and on macrophages. This part should certainly be expanded.
Even in the section where the effect of SCFA is discussed, the authors should go a little deeper by expanding the now much studied topic. Perhaps also to discuss the studies that include the direct use of SCFA as a therapy for IBD orthe formulation of probiotics specially designed to produce greater quantities of these metabolites may be interesting in this type of review.
Author Response
Reviewer 1) Comments and Suggestions for Authors
This manuscript discusses an interesting topic, such as the relationship between intestinal microbiota, epithelial cells and the immune system, with particular attention to macrophages which represent the most abundant immune population at the LP of the colon. However, some parts of the manuscript need improvement. In particular, paragraph 2 in which the authors discuss the relationship between macrophages and microbiota should be considerably expanded.
Reply, As seen in line 144 to line 197 in the text with label, we have already expanded the relationship between macrophages and microbiota.
In the discussion of this interesting topic, the authors cite various articles that also affirm partly conflicting theses sequentially confusing the reader especially if a little inexperienced. The authors should reformulate this paragraph in order to make it clearer by separating the articles concerning the anti-inflammatory effect exerted by the microbiota from those concerning the pro-inflammatory effect of the microbiota.
Reply, In this manuscript “a specific network comprised of macrophages, epithelial cells and gut microbiota for gut homeostasis”, we mainly discussed the anti-inflammatory effects of gut microbiota for gut homeostasis. To avoid confusion, we have deleted a couple of sentences, which are related to pro-inflammatory effects of the microbiota.
Furthermore, in the part concerning secondary bile acids produced by the microbiota in the literature there are now many studies also concerning specifically the polarization of macrophages and a review on this topic was published recently in this journal. Moreover, bile acid receptors are emerging as an interesting therapeutic target in IBD, being expressed both in epithelial cells and on macrophages. This part should certainly be expanded.
Reply, As seen in line 171-187 and line 260-268 in the text with label, we have already expanded these.
Even in the section where the effect of SCFA is discussed, the authors should go a little deeper by expanding the now much studied topic. Perhaps also to discuss the studies that include the direct use of SCFA as a therapy for IBD or the formulation of probiotics specially designed to produce greater quantities of these metabolites may be interesting in this type of review.
Reply, As seen in line 143-159 and line 234-249 in the text with label, we have also already expanded the effects of SCFA in section.
Reviewer 2 Report
This review is very hard to be read. It lacks a clear and consequent flow. As an example, the role of macrophage is is not completely described in the relative chapter, but is considered in various paragrephs without a clear order. Incredibly, the same title is used for section 3.1 and 4.2. Very often, a phenomenon is described without clarifying its consequence or addressing the related mechanisms (cfr. for example line 131-133, 143-145, 202-203, 205-206, 211-212 and many other). Many spelling mistakes (eg, lumenal or luminal used interchangebly; often the phrae lacks the verb or the object(cfr. line 179-180, 242-3 and many other). I stopped reading at chapetr 4.2 whose title is identical to 3.1, as already mentioned.
I suggest a complete reorganization of the paper, organized in 3 sections, i.e. macropahge, gut microbiota and gut epithelium, and maybe in sub-sections dealing with the relations between each component with one of the other two. Possibly some tables, summarizing the main findings.
Please, quote correctly the references: as an example, Ref 65 deals with children with autism spectrum disorder only.
Author Response
Reviewer 2)Comments and Suggestions for Authors
This review is very hard to be read. It lacks a clear and consequent flow. As an example, the role of macrophage is is not completely described in the relative chapter, but is considered in various paragrephs without a clear order. Incredibly, the same title is used for section 3.1 and 4.2. Very often, a phenomenon is described without clarifying its consequence or addressing the related mechanisms (cfr. for example line 131-133, 143-145, 202-203, 205-206, 211-212 and many other). Many spelling mistakes (eg, lumenal or luminal used interchangebly; often the phrae lacks the verb or the object (cfr. line 179-180, 242-3 and many other). I stopped reading at chapetr 4.2 whose title is identical to 3.1, as already mentioned.
Reply, First, there are different titles in section 3.1 and 4.2. Section 3.1 is “Gut Epithelial Structure and Function Needs Gut Microbiota”; whereas section 4.2 is “Gut Epithelial Structure and Function Needs Gut Macrophages”.
Secondly, we have further clarified some consequences or also addressed relative mechanisms.
Thirdly, we have also already corrected some spelling mistakes and sentences. Meanwhile, we have also improved the text.
I suggest a complete reorganization of the paper, organized in 3 sections, i.e. macropahge, gut microbiota and gut epithelium, and maybe in sub-sections dealing with the relations between each component with one of the other two. Possibly some tables, summarizing the main findings.
Reply, In this manuscript, we mainly addressed the relationship among gut microbiota, gut epithelial cells and macrophages, especially the relationship between gut microbiota and macrophages, gut microbiota and gut epithelial cells as well as gut epithelial cells and macrophages. Thus, it is better for present organization.
Please, quote correctly the references: as an example, Ref 65 deals with children with autism spectrum disorder only.
Reply, We have already deleted ref 65.
Reviewer 3 Report
In this manuscript, the authors summarized the cooperative and dynamic interactions among gut microbiota, gut macrophages and intestinal epithelial cells; discussed the network of macrophages, gut microbiota and epithelial cells for gut homeostasis; outlined the effects of gut macrophages and gut epithelial cells on microbiota; also discussed the mechanisms of the network of gut macrophages, epithelium and microbiota for over-reactive or suppressive immune responses in different pathogens invasion, susceptible and pathological states. This study is overall well-designed, well written, and the literatures are well interpreted and the logic discussion can be recognized as well.
This is a systematic review on a focused question of how the macrophages, intestinal epithelial cells and microbiota interact and cooperate for inflammation responses as a way of network to modulate and maintain the balance between gut microbes and the host immune system. And the authors identified the network comprised of macrophages, intestinal epithelial cells and gut microbiota plays an very important role of over-reactive or suppressive immune responses in different pathogens invasion, susceptible and pathological states. This scope is a high clinical relevance and I believe that this work will be of broad interest to researchers in the research of interactions of immune system and gut microbiota.
The major contribution of this work is that the authors systematically analyzed the mechanisms of the complex network among macrophages, intestinal epithelial cells and gut microbiota, researchers can benefit from this to better understand the interaction between immune system and gut microbiota.
It's a well written and prepared manuscript with very good readability.
The conclusions the authors present consistent with the evidences compiled from literatures, the main question posed is well addressed.
Author Response
Thanks very much for your comments.
Round 2
Reviewer 2 Report
Paper is now suitable for publication